# Effect of Organic and Chemical Fertilizer Application on Apple Nutrient Content and Orchard Soil Condition

Takamitsu Kai [1,*] and Dinesh Adhikari [2]

1 Kurokawa Field Science Center, Meiji University, Kawasaki 215-0035, Japan
2 Compass Minerals Innovation Center, Stilwell, KS 66085, USA; adhikarid@compassminerals.com
* Correspondence: kai_takamitsu@meiji.ac.jp; Tel.: +81-44-980-5300

**Abstract:** Apple cultivation in Japan typically involves the use of chemical fertilizers and pesticides which can damage the environment. Therefore, in this study we investigated the orchard soil biochemical characteristics as well as the fruit nutrient contents, and pesticide residues of apples grown either organically (organic fertilizers + reduced pesticides) or with conventional chemical fertilizers and pesticide rates. Compared with conventional chemical fertilizer treatment, the organic fertilizer treatment produced fruit with significantly higher contents of sugar, as well as soil with higher total carbon, total nitrogen, and total phosphorus. There were also significantly greater soil bacterial biomass and N circulation in the organically fertilized treatments. Minimal pesticide residues were detected in the organically fertilized fruits, but in the apples cultivated with conventional rates of fertilizers and pesticides there were significantly higher levels of propargite that was used to control spider mites. These residue levels from the conventionally fertilized orchards exceeded European and Codex residue standards. These results indicate that environmentally friendly arboricultural soil management practices, such as organic fertilizer and reduced pesticide cultivation can enhance nutrient cycling in soil, reduce the burden on the environment, and promote food safety and security.

**Keywords:** apple orchard; microorganism; soil fertility; organic fertilizer cultivation; agricultural environment; environmental protection





## 1. Introduction

Apple (*Malus domestica* Borkh.) is an important fruit crop in many countries, including Japan [1,2]. However, the average annual apple productivity in Japan is considerably lower than in other developed countries like Austria, New Zealand, and the USA [3]. The poor productivity seems to be related to poor soil fertility [1,4,5], probably because most apple production in Japan uses large amount of chemical fertilizers and pesticides. Organic production methods could help reduce the amount of chemicals that damage soils, but there are very few organic orchards in Japan [6]. However, when using organic fertilizer methods, it is also important to ensure that there are sufficient nutrients available to ensure good growth of the fruit trees. Previous studies suggest that productivity can be up to 30% lower when using organic methods [7,8]. Therefore, in order to maximize the potential of organic apple production systems it is important to improve nutrient availability as well as the abundance and activity of soil microorganisms [9].

In addition to the use of chemical fertilizers, there are three other types of fertilizer management systems that are used in apple cultivation: natural conditions (with no cultivation, fertilizers, or pesticides), the use of organic fertilizers only, and the combined use of organic and chemical fertilizers [10–12]. However, there is a need to reduce the use of chemical fertilizers and pesticides in order to reduce risks to health and the environment.

Previous studies on other crops in Japan found that organic fertilizer application improved both plant growth and soil conditions compared with chemical fertilizer application. For example, organically fertilized paddy rice had more panicles, higher grain weights, and more ripened grains, as well as less acid soils and more soil carbon [13].

Moreover, organically fertilized cherry tomatoes produced larger and heavier fruits with greater lycopene content [14]. The tomato field also had higher soil nutrient contents and greater microbial activity and biomass. In another study a vegetable mustard grown under organic fertilization growth parameters such as leaf length, shoot height, and shoot width were higher compared with chemical fertilizer management [15]. Moreover, in a study conducted on different apple orchard cultivation systems, it was found that the highest levels of carbon, nitrogen, phosphorus, and potassium were generally found in the soils of organically farmed orchards, and those orchards also had greater soil microbial activity [10].

These studies indicate that the higher soil carbon and nitrogen in organically fertilized soils can increase the bacterial biomass, leading to enhanced N and P circulation, thus producing a rich soil environment with active soil microorganisms. Productive apple orchards generally have soils richer in carbon, nitrogen, phosphorus, and potassium than annual cropland soils [16,17]. Therefore, organic fertilizer management systems that improve soil nutrient contents should be considered for orchard soils in particular [18].

The novelty about this study is that we directly compared in detail both fruit and soil parameters in the same soil type (adjacent orchards) with different treatments (organic versus conventional). In this study, we evaluated fruit and leaf nutrient contents, soil biochemistry, and pesticide residues in apple orchards cultivated in the same area using either (1) organic fertilizers and reduced pesticides, or (2) conventional chemical fertilizers and synthetic pesticides.

## 2. Materials and Methods

### 2.1. Study Sites

This study was conducted in six high-yielding apple orchards (cultivar 'Fuji') in Shiojiri City (latitude is 137.9532° N, longitude is 36.1148° E.), Nagano Prefecture, Japan. The local climate is humid temperate, and the average highest and lowest daily temperatures in 2020 were 25.9 °C in August and 1.1 °C in January, respectively [19]. The six orchards were adjacent to each other, in a flat basin at an elevation of about 710 m. Each covered an area of about 3000 $m^2$ (60 m $\times$ 50 m). The trees were about 15 years old, with 30 trees per orchard (providing 10 m $\times$ 10 m per tree). The average yield in each orchard is about 30 t $ha^{-1}$. Three of the orchards were under organic fertilizer management with reduced pesticides, and the other three were under conventional chemical fertilizer and synthetic pesticide management. The organic fertilizer orchards had not received any chemical fertilizer for 15 years. Instead, they had typically received 5.3 t $ha^{-1}$ fish meal plus pruned branches which were chipped in a wood-chipper. These materials were applied to the orchard soil surface in December each year after harvesting. The organic orchards also receive reduced pesticide doses (less than half of the conventional dose), that have been applied only twice annually for the last 15 years, mainly to control black star disease. The reduced-pesticide application method is intended to produce apples that are safe for consumption but still of reliable quality. The organic fertilizer, wood chip, and reduced pesticide applications in the year of our study were the same as described above.

The chemical fertilizer orchards typically received chemical fertilizer at an annual rate of 160 kg $ha^{-1}$-N, 50 kg $ha^{-1}$-$P_2O_5$, 120 kg $ha^{-1}$-$K_2O$, according to Nagano prefecture standard recommendations, with half the chemical fertilizer applied in December (after harvest) and half in April (before flowering). In the chemical fertilizer orchards, the pesticides are applied fifteen times each year, based on the prefectural agricultural policy [20]. These chemical fertilizer and pesticide applications in the year of our study were the same as described above. In all six orchards, the orchard floor undergrowth was typically cut five times between May and September.

### 2.2. Leaf and Fruit Nutrient Contents

Five randomly chosen trees in each orchard were sampled for leaves in July 2020 and fruit in November 2020. A composite juice sample was prepared from the fruits of the five

sampled trees, then three measurements of fruit sugar content and acidity were made on each composite sample to provide an average value. The sugar content of the undiluted juice was measured by using a sugar–acidity meter (PAL-BX/ACID5; Atago, Tokyo, Japan). To measure acidity, the undiluted juice was diluted fifty-fold by mixing with purified water (100 rpm, 5 min) and then measuring with the sugar–acidity meter. The sugar/acid ratio was calculated as sugar content divided by acidity.

### 2.3. Soil Sampling and Analysis for Physical and Chemical Properties

Soils were sampled in November 2020. Soil samples (0–100 cm) were taken from each orchard using a DIK-1640 boring stick (Daiki Rika Kogyo Co., Ltd., Saitama, Japan), to assess soil texture and density. Soil color was determined using Munsell Soil Color Charts [21]. For particle size analysis, particle sizes of 75 μm or more were obtained by sieve analysis, and particles less than 75 μm were obtained by sedimentation. Soil density was measured by the pycnometer method, using a Gay–Lussac type specific gravity bottle (Asone, Hyogo, Japan).

Composite soil samples of the top 15 cm (excluding the surface crust of 2–3 cm) were taken from around the base of five randomly selected trees for analyzing the chemical and biological properties. The composite samples were first placed in sealed plastic bags containing air for continued microbial activity. The fresh soil was used for chemical and biological analysis as soon as possible. Prior to the fresh soil analysis, the soil was sieved through a 2 mm sieve, then plant roots and stones were removed.

Some of the freshly sieved soil was also air dried for 7 days, then sieved again through a 2 mm sieve prior to analysis for physical properties.

The following analyses were conducted on the fresh soil samples. Ammonium–nitrogen ($NH_4^+$-N) and nitrate–nitrogen ($NO_3$-N) were extracted with 1 mol $L^{-1}$ KCl, followed by measurement using the indophenol blue and brucine methods, respectively [22]. Water soluble phosphorus (SP) and potassium (SK) were extracted from the soil by shaking a soil: water suspension (1:20, *w/v*) at 100 rpm for 1 h, and then analyzing the extracts by the molybdenum blue method [9] and atomic absorption spectrophotometry, respectively. Total carbon (TC) was analyzed using an SSM-5000A total organic carbon analyzer (Shimadzu, Kyoto, Japan). To measure total nitrogen (TN), phosphorus (TP) and potassium (TK) contents, soil was digested with $H_2SO_4$ and $H_2O_2$ in a Kjeldatherm digestion unit (Gerhardt, Königswinter, Germany). Then the respective values were obtained by measuring the amounts of $NH_4^+$-N, SP, and SK in the digested sample extract. The pH of the soil was measured in a soil:water suspension (1:2.5, *w/v*) with a LAQUA F-72 pH meter (Horiba Scientific, Kyoto, Japan).

### 2.4. Soil Biological Properties

Various biological properties were measured to assess the ability of the soil microbial biomass to circulate nutrients. Total bacterial biomass was estimated by quantifying environmental DNA (eDNA) [23].

Nitrogen circulation was evaluated as the soil's ability to convert organic N to $NO_3^-$ [17]. Nitrogen circulation of organic N involves various decomposition processes: proteins → peptides → amino acids → ammonium → nitrite → nitrate. These processes are carried out by various groups of microorganisms, with different bacterial groups responsible for $NH_4^+$ oxidation ($NH_4^+ \rightarrow NO_2^-$) and $NO_2^-$ oxidation ($NO_2^- \rightarrow NO_3^-$). The N circulation activity was estimated from the area of the triangular chart made by plotting the values of the $NH_4^+$ oxidation activity, $NO_2^-$ oxidation activity and total microbial biomass [24]. Larger triangles indicated more N circulation in the soil. Phosphorus circulation was estimated by incubating soil for three days and measuring the amount of soluble P released from organic phosphorus in the form of phytic acid. The P circulation score ranged from 100 if all the phytic acid was converted to soluble P without any adsorption by other minerals, to 0 if no soluble P was detected [25].

## 2.5. Pesticide Residues

We used the residual pesticide soap method [26] to assess the amount of pesticide residue in the apples. First, four whole apples, without any visible damage, were analyzed to screen for residues of approximately 400 pesticides. The pesticide residues were analyzed by supercritical fluid chromatography (SFC) using an SFC system (Waters Co., shinagawa, Tokyo, Japan). The mobile phase was $CO_2$ (99.9% grade) with an MeOH modifier containing 0.1% (*w/v*) $HCOONH_4$, a flow rate of 3 mL min$^{-1}$, an oven temperature of 35 °C and 10 MPa back pressure. Mass spectrometry was conducted using a Xevo TQ mass spectrometer (Waters Co.). Ionization was conducted by electrospray ionization (ESI), with positive polarity, 3.0 kV capillary voltage, desolvation at 600 °C with a desolvation gas flow rate of 800 L h$^{-1}$, a cone gas flow rate of 60 L h$^{-1}$ and a collision gas flow rate of 0.15 mL min$^{-1}$.

## 2.6. Statistical Analysis

Data analysis was conducted using Excel Statistics 2016 for Windows (Social Survey Research Information Co., Ltd., Tokyo, Japan). Significant differences were determined by one-way analysis of variance followed by mean comparisons by Fisher's least significant difference (LSD) test ($p < 0.05$).

## 3. Results

### 3.1. Leaf Nutrient Content

The TC, TN, TP, and TK contents of apple leaves are shown in Table 1. The TC content was significantly higher under organic fertilizer management than under chemical fertilizer management. In contrast, the organically fertilized trees had higher leaf TN than the chemically fertilized trees. However, the TP and TK contents did not differ significantly between the fertilizer management systems.

**Table 1.** The total carbon (TC), total nitrogen (TN), total phosphorus (TP), and total potassium (TK) contents of apple leaves.

| Experimental Treatments | TC (mg kg$^{-1}$) | TN (mg kg$^{-1}$) | TP (mg kg$^{-1}$) | TK (mg kg$^{-1}$) |
|---|---|---|---|---|
| Organic fertilizer | 464,667 ± 8087 [z] | 10,997 ± 150 | 1245 ± 18 | 6149 ± 332 |
| Chemical fertilizer | 485,433 ± 4879 | 9943 ± 514 | 1503 ± 146 | 7491 ± 1603 |
| LSD-test [y] | * | * | n.s. | n.s. |

[z]: Mean ± standard deviation of a sample (total C, total N, total P, total K; $n = 25$). [y]: * indicates that values in a column are significantly different at $p < 0.05$ according to the least significant difference LSD method; n.s. indicates no significant difference.

### 3.2. Fruit Nutrient Content

The TC, TN, TP, TK, and sugar and acidity contents of the apple fruits are shown in Table 2. There were no significant differences in the TC, TN, TP, TK, or acidity between the fruits in the two fertilizer treatments. However, the sugar content of fruits under organic fertilizer management was significantly higher than in fruits under chemical fertilizer management.

**Table 2.** The total carbon (TC), total nitrogen (TN), total phosphorus (TP), and total potassium (TK) of apple fruits.

| Experimental Treatments | TC (mg kg$^{-1}$) | TN (mg kg$^{-1}$) | TP (mg kg$^{-1}$) | TK (mg kg$^{-1}$) | Sugar (Brix%) | Acidity (%) | Sugar–Acidity Ratio |
|---|---|---|---|---|---|---|---|
| Organic fertilizer | 76,283 ± 9237 [z] | 518 ± 37 | 483 ± 121 | 2250 ± 842 | 16.0 ± 1.9 | 0.41 ± 0.17 | 39.0 ± 22.9 |
| Chemical fertilizer | 84,200 ± 16,985 | 480 ± 28 | 367 ± 124 | 2833 ± 205 | 14.5 ± 2.0 | 0.38 ± 0.26 | 38.2 ± 17.8 |
| LSD-test [y] | n.s. | n.s. | n.s. | n.s. | ** | n.s. | n.s. |

[z]: Mean ± sample standard deviation (total C, total N, total P, total K: $n = 5$, sugar, acidity, sugar-acidity ratio: $n = 140–148$). [y]: ** indicates that values in a column are significantly different at $p < 0.05$ according to the LSD method; n.s. indicates no significant difference.

### 3.3. Soil Physical and Chemical Properties

The surface and subsurface soil textures to a depth of 100 cm in six orchards were clay loam (CL). The soil particle densities were 2.47 g cm$^{-3}$ in six orchards.

The soil chemical properties are shown in Table 3. TC, TN, TP, and TK were all significantly higher in the organic fertilizer orchard soils than in the conventional fertilizer soil. The C/N ratio was also significantly higher in the soils under organic fertilizer management. However, there were no significant differences in NO$_3^-$-N, NH$_4^+$-N, soluble P, soluble K contents, and EC between the two orchard fertilizer systems. The pH was higher in the chemically fertilized soil than in the organically fertilized soil.

**Table 3.** Chemical properties of soils of orchards under organic and chemical fertilizer management systems.

| Experimental Treatments | TC (mg kg$^{-1}$) | TN (mg kg$^{-1}$) | TP (mg kg$^{-1}$) | TK (mg kg$^{-1}$) | C/N Ratio (%) | NO$_3^-$-N (mg kg$^{-1}$) | NH$_4^+$-N (mg kg$^{-1}$) | Soluble P (mg kg$^{-1}$) | Soluble K (mg kg$^{-1}$) | pH | EC (mS cm$^{-1}$) |
|---|---|---|---|---|---|---|---|---|---|---|---|
| Organic fertilizer | 63,917 ± 14,698 [z] | 5906 ± 1658 | 19,368 ± 5090 | 5011 ± 535 | 10.8 ± 0.7 | 66.0 ± 27.4 | 1.7 ± 0.5 | 222 ± 76 | 413 ± 141 | 6.9 ± 0.1 | 0.33 ± 0.12 |
| Chemical fertilizer | 18,447 ± 5227 | 911 ± 262 | 6132 ± 397 | 6789 ± 101 | 20.2 ± 1.4 | 19.0 ± 25.5 | 6.0 ± 4.1 | 114 ± 37 | 181 ± 55 | 7.6 ± 0.2 | 0.39 ± 0.39 |
| LSD-test [y] | * | * | * | * | ** | n.s. | n.s. | n.s. | n.s. | * | n.s. |

[z]: Mean ± sample standard deviation (total C, total N, total P, total K, C/N ratio, NO$_3^-$-N, NH$_4^+$-N, soluble P, soluble K, pH, and EC; *n* = 5). [y]: * and ** indicate that values in a column are significantly at *p* < 0.05 and *p* < 0.01, respectively, according to the LSD method; n.s. indicates no significant difference.

### 3.4. Soil Biological Properties

The biological properties of the soil are shown in Table 4. Bacterial biomass, NH$_4^+$ oxidation activity, and N circulation activity were 3–7 times higher under organic fertilizer management than under chemical fertilizer management. However, there were no significant differences between the two fertilizer systems for NO$_2^-$ oxidation activity and P circulation activity.

**Table 4.** Biological properties of soils of orchards under organic and chemical fertilizer management systems.

| Experimental Treatments | Bacterial Biomass (×10$^8$ Cells g$^{-1}$) | NH$_4^+$ Oxidation Activity (Point) | NO$_2^-$ Oxidation Activity (Point) | N Circulation Activity (Point) | P Circulation Activity (Point) |
|---|---|---|---|---|---|
| Organic fertilizer | 23.6 ± 1.7 [z] | 95.7 ± 6.1 | 99.8 ± 0.2 | 97.0 ± 4.2 | 16.0 ± 3.7 |
| Chemical fertilizer | 4.0 ± 3.2 | 32.3 ± 9.2 | 37.3 ± 44.6 | 13.7 ± 7.6 | 8.7 ± 7.4 |
| LSD-test [y] | ** | ** | n.s. | ** | n.s. |

[z]: Mean ± sample standard deviation (bacterial biomass, NH$_4^+$-N oxidation activity, NO$_2^-$-N oxidation activity, N circulation activity, and P circulation activity; *n* = 5). [y]: ** indicate that values in a column are significantly different at *p* < 0.01, according to the LSD method; n.s. indicates no significant difference.

### 3.5. Pesticide Residues

The fruit pesticide residues are shown in Table 5, together with the Japanese, European Union (EU), international, Chinese, and Korean pesticide residue standards. Propargite is an herbicide, acetamiprid and acrinathrin are insecticides, and Trifloxystrobin and Boscalid are fungicides [27]. Pesticide residues in the organic fertilizer treatment were undetectable or minimal and did not exceed any of the safety standards. Residues of propargite and trifloxystrobin were significantly higher in the chemical fertilizer treatment than in the organic fertilizer treatment. Moreover, even though the propargite residues did not exceed the Japanese Food Sanitation Law, Chinese, or Korean residue standards, they did exceed the EU and international (codex) standards.

**Table 5.** Fruit pesticide residues and Japan, European Union (EU), international (Codex), Chinese, and Korean pesticide residue standards.

| Experimental Treatments | Propargite (mg kg$^{-1}$) | Trifloxystrobin (mg kg$^{-1}$) | Acetamiprid (mg kg$^{-1}$) | Boscalid (mg kg$^{-1}$) | Acrinathrin (mg kg$^{-1}$) |
|---|---|---|---|---|---|
| Organic fertilizer | N.D ± 0.00 [z] | 0.01 ± 0.00 | 0.02 ± 0.00 | N.D ± 0.00 | N.D ± 0.00 |
| Chemical fertilizer | 0.37 ± 0.01 | 0.07 ± 0.01 | 0.03 ± 0.00 | 0.01 ± 0.00 | 0.01 ± 0.00 |
| LSD-test [y] | * | * | n.s. | n.s. | n.s. |
| Residual pesticide standards | | | | | |
| Japan (mg kg$^{-1}$) | <5 | <3 | <2 | <2 | <0.7 |
| EU (mg kg$^{-1}$) | <0.01 | <0.7 | <0.4 | <2 | <0.02 |

**Table 5.** *Cont.*

| Experimental Treatments | Propargite (mg kg$^{-1}$) | Trifloxystrobin (mg kg$^{-1}$) | Acetamiprid (mg kg$^{-1}$) | Boscalid (mg kg$^{-1}$) | Acrinathrin (mg kg$^{-1}$) |
|---|---|---|---|---|---|
| Codex (mg kg$^{-1}$) | <3 | <0.7 | <0.8 | <2 | - |
| China | <5 | <0.7 | <0.8 | <2 | - |
| Korea | <5.0 | <0.7 | <0.3 | <1.0 | 0.5 |

$^z$: Mean ± sample standard deviation (propargite, trifloxystrobin, acetamiprid, boscalid, acrinathrin; *n* = 5). $^y$: * indicates that values in a column are significantly different at *p* < 0.05, according to the LSD method; n.s. indicates no significant difference.

## 4. Discussion

This study compared the effects of two fertilizers and pesticide management systems on apple nutrient contents and soil conditions. The average nitrogen and phosphorus circulation activities of the orchard soils are shown in Figures 1 and 2, respectively. The organically fertilized soils with reduced pesticides had greater nutrient circulation activities than in the soils receiving conventional chemical fertilizer and synthetic pesticides. These results indicate larger amounts of nitrogen and phosphorus cycled in the organically managed fields. Nutrient cycling is directly related to soil microbial community and organic matter contents [28]. Quality of organic matter is also important for higher microbial biomass and diversity in soil [29]. Our study also indicates the presence of high quality organic matter and richer microbial community in the orchards under organic management.

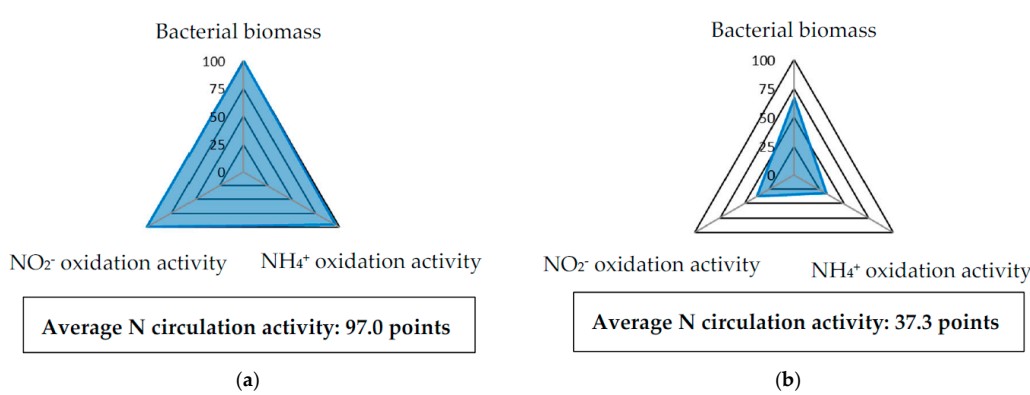

**Figure 1.** Average nitrogen circulation activity in (**a**) organically managed soils and (**b**) conventionally managed soils.

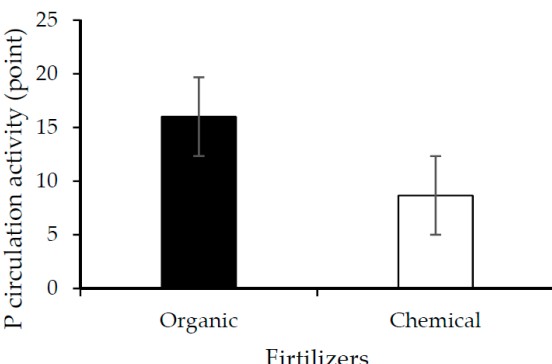

**Figure 2.** P circulation activity in organically and chemically fertilized soils.

In orchard and other upland agricultural soils, the minimum recommended C and N contents for achieving a high level of nutrient circulation activities are 25,000 and 1500 mg kg$^{-1}$, respectively, and the recommended C/N ratio is 10–25 [16,30,31]. In the current study, these recommended values were achieved in the orchard soil under organic fertilization, but not in the soil under chemical fertilizer and pesticide management (Figure 3). The total carbon and total nitrogen in the organic soil remained above the recom-

mended values, even after November harvest. However, in the chemically fertilized soil, chemical fertilizer was applied in December and April, which may have provided plentiful nutrients during the spring and summer, but there may have been reduced availability by the November harvest season. The low carbon in the chemically fertilized soil was probably because this management provided little organic materials. Higher amounts of organic matter not only contributes for greater microbial biomass but also for reduced runoff, erosion and leaching of nutrients [32]. Moreover, organic matter in soil acts like a bank by storing the nutrients during the high inputs and releasing during the periods of low input leading to a steady supply of nutrients to the plants [33].

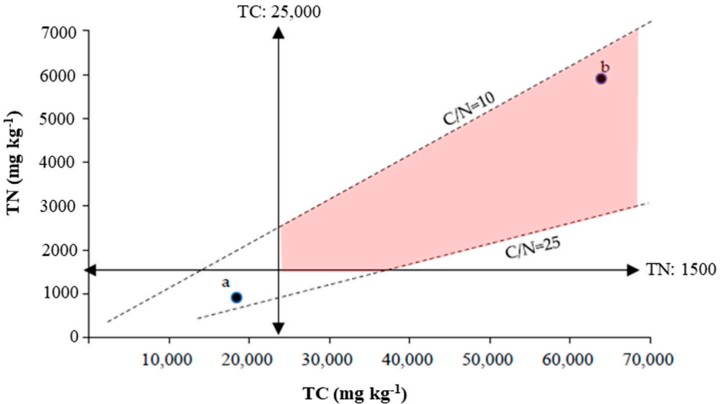

**Figure 3.** Values of total C and total N in (a) the organic, (b) the chemical fertilizer treatments (*n* = 2). Light red zone corresponds to the recommended values of TC, TN, and C/N ratio for achieving a high level of nutrient circulation activities [16,28,29].

The average total carbon and bacterial biomass in various kinds of orchard soils are shown in Figure 4 [10,16,34]. In the current study, the organically fertilized soil had greater total carbon and bacterial biomass than the convention chemically fertilized soil. This suggests that the chemical fertilizers and synthetic pesticides reduced the soil bacterial biomass. In order to characterize orchard soils, we analyzed the mean bacterial biomass and TC values separately and characterized four soil groups (Figure 4). The group 1 soils had high TC and microbial biomass values suggesting that these soils probably received organic fertilizer and reduced pesticide application. In contrast, the low bacterial biomass and TC content in the group 4 soils suggest that they probably received more chemical fertilizers and pesticides that directly affected the soil microorganisms. Where soil microbial activity is low, it is likely that the soil has regularly received chemical fertilizers and pesticides.

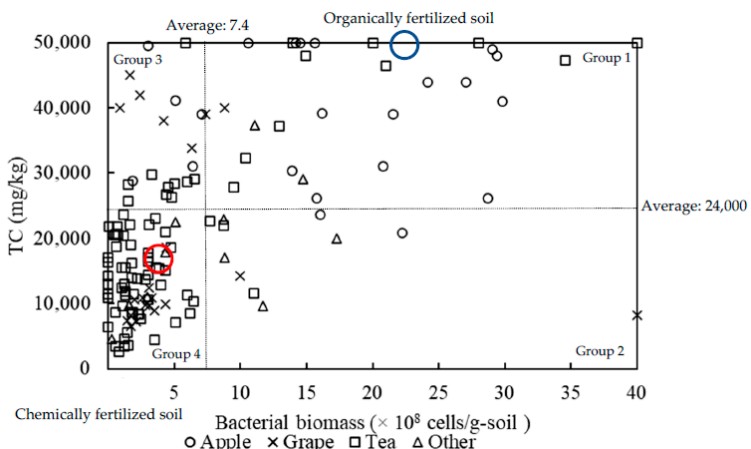

**Figure 4.** Relationship between total carbon (TC) and bacterial biomass in orchard soils and in the (○) organic and (●) conventional farming systems in the current study (*n* = 2).

A nationwide field survey in Japan reported that paddy fields that had received organic fertilizer and reduced pesticides had a higher plant biomass and higher number of aquatic animals than fields that had been subjected to conventional farming methods [35]. Furthermore, the report also showed that management crop rotation could lead to increases in population of some species. Therefore, organic farming and reduced-pesticide application can contribute to conserving biodiversity and maintaining biological growth and habitat. Our current study also reveals that environmentally friendly management of orchard soil, using organic farming techniques and reduced-pesticide application, enhances nutrient recycling and improves fruit quality. This in turn reduces the burden on the environment, thereby improving food safety and security.

Countries have safety standards to ensure that pesticide residues in food do not cause harm to humans. In Japan, the Ministry of Health, Labour, and Welfare has established residue standards for pesticides, feed additives, and veterinary drugs [31]. Residue levels are set for each food product, within limits assessed by the Food Safety Commission as safe for human consumption. Other countries and regions also set their own residue standards, sometimes stricter and sometimes less strict than in Japan. In the current study, the level of propargite residue in the conventionally farmed apples was within the Japanese standard for apples but did not meet European Union and International standards. This means that it may while such conventionally cultivated apples can be sold in Japan, export sales could be limited. However, there was no such problem with the organically cultivated apples.

This study also revealed that the different soil conditions in the two orchard management systems affected the sugar content, acidity, and sugar–acidity ratio of the harvested apples. Soil fertility can be improved and maintained through using appropriate organic fertilizers. The agricultural products grown in such orchards can be expected to grow healthily and improve their yield because of the continuous supply of nutrients [18].

Therefore, organic fertilizers can increase soil nutrient parameters like total carbon, nitrogen, and C/N ratio, which in turn can increase soil microbial biomass and nutrient circulation activities in the soil. This suggests that organic farming methods could be used to improve food safety and soil conservation in apple orchards while still maintaining a good productivity.

### 5. Conclusions

The results of this study show that the organically fertilized apple orchards had higher soil carbon, nitrogen, phosphorus, and microbial biomass, leading to more intensive soil nitrogen cycling. Moreover, the apples grown in organic orchards had minimal pesticide residues. These results suggest that appropriate organic management can ensure sufficient nutrient availability to obtain a good productivity with a reduced pesticide residue levels.

**Author Contributions:** T.K. contributed to conceptualization, methodology and investigation, D.A. contributed to methodology, T.K. and D.A. contributed to manuscript writing and editing. Both authors have read and agreed to the published version of the manuscript.

**Funding:** This work was supported by JSPS KAKENHI Grant Number JP19K15937.

**Institutional Review Board Statement:** Not applicable.

**Informed Consent Statement:** Not applicable.

**Data Availability Statement:** Not applicable.

**Conflicts of Interest:** The authors declare no conflict of interest.

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
