# Peer review of "Effect of Organic and Chemical Fertilizer Application on Apple Nutrient Content and Orchard Soil Condition"

_agriculture, doi:10.3390/agriculture11040340_

Round 1
Reviewer 1 Report
This manuscript deals with the effect of organic/chemical fertilizer on nutrient content in apple and soil condition.
Line 43. How about the effect of soil physical properties on apple nutrient and productivity?
Line 89-92. To describe the aim of this study, please emphasize the originality and novelty of this article.
Line 96. Describe latitude and longitude.
Line 137. How did you prepared soil for analysis? air dried? using dry-oven? sieve number?
2.4. If possible, further study determined the analysis item such as soil enzyme activity.
3.3 and 3.4 seems to be incorporated in the one sub-section number.
Table 4. Isn't there no unit in soil enzyme item activity?
Figure 3 is very interesting results.
Figure 4 is hard to read sufficiently. Please modify it more easy to reader.
Line 338-343. How about adding a discussion it with the case of China or Korea, where the climate is similar because it is far closer than the United States or Europe.
Conclusion. It is hard to agree as conclusion. It has to be completely rewritten.
Author Response
Reviewer(s)' Comments to Author:
Reviewer: 1
We thank Reviewer: 1 for taking time to review this paper and giving us constructive comments. We have carefully read all the comments, responded to them, and made revisions according to the suggestions.
Comments to the Author
1. Line 43. How about the effect of soil physical properties on apple nutrient and productivity?
There are few studies/results from other researchers that show the influence of physical properties of soil. Research examples are few and far between. (Page 1, Line 41- 42)
2. Line 89-92. To describe the aim of this study, please emphasize the originality and novelty of this article.
We have added descriptions to make our understanding clearer. “The original / novel point about our study is that we directly compare in detail both fruit and soil parameters in the same soil type (adjacent orchards) with different treatments (organic versus conventional). “(Page 2, 61-63)
3. Line 96. Describe latitude and longitude.
Thank you for your comment. We have added the latitude and longitude. “Latitude is 137.9532, longitude is 36.1148.” (Page 2, 70)
4. Line 137. How did you prepare soil for analysis? Air dried? Using dry-oven? Sieve number?
Thank you for your comment. We have added the following description to make the understanding clearer. “The composite samples were first placed in sealed plastic bags containing air for continued microbial activity. The fresh soil was used for chemical and biological analysis that was conducted as soon as possible after sampling. Prior to the fresh soil analysis, the soil was sieved through a 2 mm sieve, then plant roots and stones were removed.
Some of the freshly sieved soil was also air dried for 7 days, then sieved again through a 2 mm sieve prior to analysis for physic. ” (Page 3, 120-121)
5. 2.4 If possible, further study determined the analysis item such as soil enzyme activity.
Thank you for your comment. It would be interesting to also have information about soil enzyme activity in the orchards, but unfortunately this analysis was not conducted in this study.
6. 3.3 and 3.4 seems to be incorporated in the one sub-section number.
Thank you for your comment. We have merged sub-sections 3.3 and 3.4 together to make a new subsection “3.3 Soil Physical and Chemical Properties” (Page 5, 187). Then, we made new sub-sections “3.4 Soil Biological Properties” (Page 5, 213) and “3.5 Pesticide Residues” (Page 6, 228).
7. Table 4. Isn’t there no unit in soil enzyme item activity?
Thank you for your comment. It is known that nitrogenous organic compounds such as protein in soil is decomposed by soil microorganisms into ammonium (NH4+), nitrite (NO2-), and nitrate (NO3-) after the low-molecular-weight transformation from protein to peptide to amino acid. In these processes, we measured ammonia oxidation activity (NH4+ → NO2-), nitrite oxidation activity (NO2- → NO3-), and the number of microorganisms. We quantified the values of ammonia oxidation activity, nitrite oxidation activity, and microbial count on the triangular radar charts to evaluate the ability of soil to convert nitrogenous organic matter to nitrate-nitrogen as the "nitrogen cycling activity evaluation value. The larger the area of the triangle, the more active the nitrogen cycle is in the soil; conversely, the smaller the area of the triangle, the lower the number of microorganisms and the less advanced the nitrogen cycling. In addition, in order for plants to be able to absorb phosphorus, it must be decomposed from phytic acid (a dominant organic form of phosphorus in soil) to plant available inorganic phosphorus (phosphorus mineralization activity). Therefore, we evaluated the ability to convert the organic form of phosphorus, called phytic acid, as the "phosphorus cycling activity” evaluation value.
8. Figure 3 is very interesting results.
Thank you very much for your comment.
9. Figure 4 in hard to read sufficiently. Please modify it more easy to reader.
Thank you very much for your comment. We have modified the markings to make them easier to read. (Page 9, 311-312).
10. Line 338-343. How about adding a discussion it with the case of China or Korea, where the climate is similar because it is far closer than the United States or Europe.
We have added some discussion about the pesticide residue levels for China and Korea. “The fruit pesticide residues are shown in Table 5, together with the Japanese, European Union (EU), Chinese, Korean and international pesticide residue standards. Propargite is a herbicide, acetamiprid and acrinathrin are insecticides, and Trifloxystrobin and Boscalid are fungicides [29]. Pesticide residues in the organic fertilizer treatment were undetectable or minimal, and did not exceed any of the safety standards. Residues of propargite and trifloxystrobin were significantly higher in the chemical fertilizer treatment than in the organic fertilizer treatment. Also, while the propargite residues did not exceed the Japanese Food Sanitation Law, nor the Chinese or Korean residue standards, they did exceed the EU and international (codex) standards.” (Page 6, 229-237)
We have revised Table 5 and the title to include the pesticide residue standard values for China and Korea. “Table 5. Fruit pesticide residues and Japan, European Union (EU), international pesticide residue standards, Chinese, and Korean residue standards.” (Page 6 239-240)
11. Conclusion. It is hard to agree as conclusion. It has to be completely rewritten.
Thank you for your comment. We have added the following descriptions to make the understanding clearer. “The results of the study show that the organically fertilized orchards had higher soil carbon, nitrogen, phosphorus and microbial biomass, leading to greater soil nitrogen cycling. Also, the organic orchards had minimal pesticide residues. These results suggest that appropriate organic management can ensure sufficient nutrient availability to obtain good apple productivity a reduced pesticide residue levels.” (Page 10, 328-334)
Reviewer 2 Report
Review Agriculture – 1156971
Effect of organic and chemical fertilizer application on apple nutrient content and orchard soil condition
The work discusses the benefits of running an extensive orchard according to ecological principles. The results largely confirm the achievements of other researchers, but due to the large number of measurements, it allowed to collect comprehensively interesting results.
In the whole work too often used article "the" for example, in the title under Table 3, or below e.g. organically, conventionally.
Detailed remarks:
Line 13-15: The sentence is not clear, needs to be rewritten. Now is that organic fertiliser produced carbon etc.
Line 18: is → were
Line 37: amounts → amount
Line 88: that → those
Line:89: evaluate → evaluated
Line 95: variety is used for taxonomic units naturally formed in nature, in the case of man-made by breeding should use the term cultivar, please check throughout the work and corect
Line 101: under → were under
Line 104: What kind of is fish meal was applied?
Line 105: soils → soil
Line 101-104: How this organic fertilisation was applied to soil?
Line 106-107: What kind of pesticide has been applied? Against what and why only twice a year? There were no any pests and diseases I other time of the year?
Line 108: cultivation → application
Line 109-111: What was the reason with such high doses of fertilization? Is this due to soil analysis or are these standard recommendations in Japan?
Line 109-115: Please use the past tense!
Line 112: Above you mentioned the use of pesticides twice a year!
Line 114: the underbrush is in the forest
Line 118, 137: skip “composite”
Line 122: onto → using or on
Line 122-123 and 125-127: the same information
Line 138: bases → base
Line 143: followed → following
Line 147, 173-174, 229-230, 290: soils → singular
Line 200-202: Please omit all the values in the text they are below in the table.
Line 214-215 The sentence needs to be rewritten
Line 223: sub-surface → subsurface
Line 232: With regard to → Regarding
Line 263, 77: Also → moreover, or in addition or furthermore and while the → the
Line 273: fertilizer → plural
Line 278: applied → were applied
Line 304: grape→ grapes
Line 308: by the use of → using
Line 318: receiving → received
Author Response
Reviewer: 2
To Reviewer:2
We thank Reviewer: 2 for taking time to review this paper. We very much and thank you for giving us constructive comments. We have carefully read all the comments, responded to them, and made revisions according to the suggestions.
The work discusses the benefits of running an extensive orchard according to ecological principles. The results largely confirm the achievements of other researchers, but due to the large number of measurements, it allowed to collect comprehensively interesting results.
Thank you for your comment.
In the whole work too often used article "the" for example, in the title under Table 3, or below e.g. organically, conventionally.
Thank you for your comment. We have asked a native English speaker to check the revised manuscript.
Line 13-15: The sentence is not clear, needs to be rewritten. Now is that organic fertiliser produced carbon etc.
Thank you for your comment. We have added the following descriptions to make the understanding clearer. “Compared with conventional chemical fertilizer treatment, the organic fertilizer treatment produced fruit with significantly higher contents of sugar, as well as soil with higher total carbon, total nitrogen, and total phosphorus.”
Line 18: is → were
Thank you for your comment. We made the correction. (Page 1. 17)
Line 37: amounts → amount
Thank you for your comment. We made the correction. (Page 1. 30)
Line 88: that → those
Thank you for your comment. We have deleted this sentence in the revised manuscript.”
Line:89: evaluate → evaluated
Thank you for your comment. We made the correction. (Page 2. 63)
Line 95: variety is used for taxonomic units naturally formed in nature, in the case of man-made by breeding should use the term cultivar, please check throughout the work and correct
Thank you for your comment. We have changed the wording to ‘cultivar’. (Page 2. 69)
Line 101: under → were under
Thank you for your comment. We made the correction. (Page 2. 76)
Line 104: What kind of fish meal was applied?
Thank you for your comment. We made the following revision to the text. “The organic fertilizer orchards have not received any chemical fertilizer for 15 years. Instead, they had typically received 5.3 t ha−1 fish meal plus pruned branches which were chipped in a wood-chipper. There materials were applied to the orchard soil surface in December each year after harvesting.” (Page 2, 7881)
Line 105: soils → soil
Thank you for your comment. We made the correction. (Page 3. 107)
Line 101-104: How this organic fertilisation was applied to soil?
Thank you for your comment. We made revised the text to explain this. “The organic fertilizer orchards have not received any chemical fertilizer for 15 years. Instead, they had typically received 5.3 t ha−1 fish meal plus pruned branches which were chipped in a wood-chipper. There materials were applied to the orchard soil surface in December each year after harvesting.” (Page 2, 78-81)
Line 106-107: What kind of pesticide has been applied? Against what and why only twice a year? There were no any pests and diseases I other time of the year?
Thank you for your comment. Although the growers surveyed may have had pests and diseases at other times of the year, they only spray synthetic pesticides twice a year (to prevent black star disease) in order to improve food safety. The apple growers surveyed in this study plan plan to obtain JAS organic certification for their apples in the future. We have revised the text to, “The organic orchards also receive reduced pesticide doses (less than half of the conventional dose), that have been applied only twice annually for the last 15 years, mainly to control black star disease..” (Page 2 81-84)
Line 108: cultivation → application
Thank you for your comment. We made the correction. (Page 2. 84)
Line 109-111: What was the reason with such high doses of fertilization? Is this due to soil analysis or are these standard recommendations in Japan?
Thank you for your comment. This is the recommended amount of fertilizer for apple cultivation in Nagano Prefecture, Japan. We have revised the text to explain this. (Page 2. 88-91)
Line 109-115: Please use the past tense!
Thank you for your comment. We revised the tense as follows: “The chemical fertilizer orchards received chemical fertilizer at an annual rate of 160 kg ha-1-N, 50 kg ha-1- P2O5, 120 kg ha-1- K2O, with half the chemical fertilizer applied in December (after harvest) and half in April (before flowering). The pesticides were applied fifteen times each year, based on the prefectural agricultural policy [21]. In all six orchards, the underbrush was typically cut five times between May and September.” (Page 2. 87-94)
Line 112: Above you mentioned the use of pesticides twice a year!
Thank you for your comment. This sentence was referring to the chemically fertilized orchards only. The ‘twice per year pesticide applications’ refers to the organic orchards. We have revised the description of the fertilizer and pesticide applications in both orchard types to make the correct meaning clearer. (Page 2. 79-94)
Line 114: the underbrush is in the forest
Thank you for your comment. We have changed the wording to ‘orchard floor undergrowth’. (Page 2. 94)
Line 118, 137: skip “composite”
Thank you for your comment, but we think that it is important to clearly tell the readers that we made one mixed sample. We think that the word ‘composite’ is a standard description for this kind of description.
Line 122: onto → using or on
Thank you for your comment. We have revised the text to, “The sugar content of the undiluted juice was measured by using a sugar–acidity meter (PAL-BX/ACID5; Atago, Tokyo, Japan).” (Page 2, 100-102)
Line 122-123 and 125-127: the same information
Thank you for your comment. We revised the text to, “The sugar content of the undiluted juice was measured by using a sugar–acidity meter (PAL-BX/ACID5; Atago, Tokyo, Japan). To measure acidity, the undiluted juice was diluted fifty-fold by mixing with purified water (100 rpm, 5 min) and then measuring with the sugar–acidity meter sensor.” (Page 2, 101-Page 3, 104)
Line 138: bases → base
Thank you for your comment. We have revised the text as suggested. (Page 3. 115)
Line 143: followed → following
Thank you for your comment. We have revised this section of the manuscript. In the revised section we think that ‘followed’ is the correct wording. (Page 3. 122)
Line 147, 173-174, 229-230, 290: soils → singular
Thank you for your comment. We have revised the words ‘soils’ to ‘soil’.
Line 200-202: Please omit all the values in the text they are below in the table.
Thank you for your comment. We have revised this section to remove the values from the text. (Page 4, 173-177)
Line 214-215 The sentence needs to be rewritten
Thank you for pointing out the poor sentence. We have revised the sentence to show the meaning we intended.
Line 223: sub-surface → subsurface
Thank you for your comment. We have revise the wording. (Page 5, 188)
Line 232: With regard to → Regarding
Thank you for your comment. We have revised the text in a way that no longer uses either wording. “The pH was higher in the chemically fertilized soil than in the organically fertilized soil. “(Page 5, 195196)
Line 263: Also → moreover, or in addition or furthermore and while the → the
Thank you for your comment. We have revised the word ‘Also’ to ‘Moreover’. However, after this the sentence requires a phrase to highlight the fact that the residue levels passed some standards (Japan, China, Korea) but failed others (EU & international). To make this clearer, we have change ‘while the’ to ‘even though the’. “Moreover, even though the propargite residues did not exceed the Japanese Food Sanitation Law, Chinese, or Korean residue standards, they did exceed the EU and international (codex) standards.”
Line 273: fertilizer → plural
Thank you for your comment. We made the correction. (Page 6, 246)
Line 278: applied → were applied
Thank you for your comment. We have revised the test (Page 6, 250)
Line 304: grape→ grapes
Thank you for your comment. We have revised this sentence and now it does not include the word grape. “The average soil total carbon and bacterial biomass in various kinds of orchard soils is shown in Figure 4.” (Page 8, 279-280)
Line 308: by the use of → using
Thank you for your comment. We have revised this section to, “In the current study, the organically fertilized soil had greater total carbon and bacterial biomass than the convention chemically fertilized soil. This suggests that the chemical fertilizers and synthetic pesticides reduced the soil bacterial biomass.” (Page 8, 279-282)
Line 318: receiving → received
Thank you for your comment. We have changed ‘receiving’ to ‘received’ and also changed other parts of the sentence to match the past tense. “A nationwide field survey in Japan reported that paddy fields that had received organic fertilizer and reduced pesticides had a higher biomass of plants and animals than fields that had been subject to conventional farming methods.” (Page 8,291-293)
Round 2
Reviewer 1 Report
Suggestions are well revised and reflectedAuthor Response
Thank!